Widespread mitovirus sequences in plant genomes

Bruenn Jeremy A. cambruen@buffalo.edu
Warner Benjamin E.
Yerramsetty Pradeep
Department of Biological Sciences, State University of New York at Buffalo , Buffalo, NY , USA
Patnaik Santosh
Electronic publication date: 2015 Apr 9
Publication date: 2015
Volume: 3
Electronic Location ID: e876
Received 2015 Feb 13; Accepted 2015 Mar 13
Copyright: © 2015 Bruenn et al.
Copyright year: 2015
Copyright holder: Bruenn et al.
License: This is an open access article distributed under the terms of the Creative Commons Attribution License, which permits unrestricted use, distribution, reproduction and adaptation in any medium and for any purpose provided that it is properly attributed. For attribution, the original author(s), title, publication source (PeerJ) and either DOI or URL of the article must be cited.
License URL: https://creativecommons.org/licenses/by/4.0/

Keywords: Narnavirus, Mycovirus, Plant mitochondria, NERVE, RNA virus

Funding: University at Buffalo Material support was provided by the University at Buffalo. The funders had no role in study design, data collection and analysis, decision to publish, or preparation of the manuscript.

==============================
The exploration of the evolution of RNA viruses has been aided recently by the discovery of copies of fragments or complete genomes of non-retroviral RNA viruses (Non-retroviral Endogenous RNA Viral Elements, or NERVEs) in many eukaryotic nuclear genomes. Among the most prominent NERVEs are partial copies of the RNA dependent RNA polymerase (RdRP) of the mitoviruses in plant mitochondrial genomes. Mitoviruses are in the family Narnaviridae, which are the simplest viruses, encoding only a single protein (the RdRP) in their unencapsidated viral plus strand. Narnaviruses are known only in fungi, and the origin of plant mitochondrial mitovirus NERVEs appears to be horizontal transfer from plant pathogenic fungi. At least one mitochondrial mitovirus NERVE, but not its nuclear copy, is expressed.

Introduction

The Narnaviridae is one of three known RNA virus families in which the viral genome is not encapsidated in a protein coat. The Narnaviridae (Esteban et al., 1994; Esteban, Rodriguez-Cousino & Esteban, 1992; Garcia-Cuellar et al., 1995; Hillman & Cai, 2013; Hong et al., 1998; Osaki et al., 2005; Rodriguez-Cousino, Esteban & Esteban, 1991; Rodriguez-Cousino & Esteban, 1992) and the Hypoviridae (Koonin et al., 1991) are presently known exclusively in fungi and the Endoviridae in plants (Fukuhara et al., 2006; Gibbs et al., 2000). All are related to families of encapsidated RNA viruses and have been proposed to be derived from common ancestors via loss of capsid polypeptide genes (Koonin & Dolja, 2013; Roossinck, 2010). As judged by the sequences of their RNA-dependent RNA polymerases (RdRPs), the closest relatives of the Narnaviridae are the leviviruses of bacteria (Garcia-Cuellar, Esteban & Fujimura, 1997) and the ourmiaviruses of plants (Rastgou et al., 2009). All of the known Narnaviridae of plant pathogenic fungi are mitoviruses: unencapsidated plus strand RNA viruses infecting mitochondria (Hillman & Cai, 2013). There may be Narnaviridae of other organisms, but possible contamination with fungi has so far prevented their identification in insects (Cook et al., 2013), although they have been identified in oomycetes (Cai et al., 2012), which have similarities to both fungi and plants. Narnaviridae have not been identified in plants (Al Rwahnih et al., 2011). The association of Narnaviridae with mitochondria is suggestive, given that their closest relatives, the leviviruses, are viruses of bacteria and the origin of mitochondria is from bacterial endosymbionts (Raven, 1970). However, rather than populating the mitochondria of all eukaryotes, the Narnaviridae have only been found in fungi. The Narnaviridae are the simplest viruses, with a single RNA segment encoding a single protein, the RdRP.

The exploration of the evolution of RNA viruses has been aided recently by the discovery of copies of fragments or complete genomes of non-retroviral RNA viruses (Non-retroviral Endogenous RNA Viral Elements, or NERVEs) in many eukaryotic nuclear genomes. (Ballinger et al., 2013; Ballinger, Bruenn & Taylor, 2012; Crochu et al., 2004; Cui & Holmes, 2012; Horie et al., 2010; Katzourakis & Gifford, 2010; Taylor & Bruenn, 2009; Taylor et al., 2011; Taylor, Leach & Bruenn, 2010). The Narnaviridae do have paleovirus sequences in eukaryotic genomes. However, these are not in fungal genomes, where they might be expected, but in plant genomes (Hong et al., 1998; Marienfeld et al., 1997). These mitovirus NERVEs are prominent in plant mitochondria, where they cannot be mistaken for contaminating sequences, because they are present in complete, relatively small, circular genomes. We examined a large collection of these mitovirus NERVEs to explore their origin, which has been ambiguous (Shackelton & Holmes, 2008). It has been suggested that the mitoviruses arose from plant genetic elements (Shackelton & Holmes, 2008). However, our results suggest the opposite—one or more integration events of a fungal mitovirus cDNA in the mitochondrial DNA of vascular plants.

Materials and Methods

Bioinformatics

Predicted amino acid sequences of NERVEs were obtained using the BLAST (Altschul et al., 1990; Altschul et al., 1997) tblastn algorithm with a number of mitovirus RdRp query sequences and the NCBI (National Center for Biotechnology Information, Bethesda, Maryland, USA) Viridiplantae subset of the refseq DNA sequence database. NERVEs were identified as having significant E values (usually less than 10−5) and/or as preserving the known conserved motifs (Bruenn, 2003) in the RdRp. We found 175 mitovirus NERVEs in the genomes of completely sequenced mitochondria. Sequences were aligned using MAFFT (Katoh et al., 2002) with the E-INS-I algorithm, a BLOSUM62 matrix, and a gap penalty of 1.53 implemented in Geneious (Kearse et al., 2012). Maximum likelihood phylogenetic analysis was carried out with PhyML 3.0 as implemented in Seaview 4.3.5 (Anisimova & Gascuel, 2006) using the model optimization of Prottest (Abascal, Zardoya & Posada, 2005), which prescribed the VT + invariable sites parameter (I) + gamma parameter for among-site variation (G)+ empirically-determined amino acid frequencies (F). Given that the Leviviridae are most closely related to the Narnaviridae (Shackelton & Holmes, 2008) (generally about 24% identity, E value of 0.23), two representative levivirus sequences were used to root the tree. Successful alignment of all of the conserved motifs of the RdRp is indicative of an accurate alignment. When alignments were evaluated with TCS (Chang, Di Tommaso & Notredame, 2014) and scores less than 8 filtered out (a very conservative choice), the position of the mitovirus NERVEs within the mitoviruses (Fig. 4) was unaltered. Synteny was assessed and visualized using the CoGe server (https://genomevolution.org/CoGe/) as described (Lyons & Freeling, 2008).

RNA and DNA isolation

To examine the possible functional significance of mitovirus NERVEs, the cellular expression of a representative mitovirus NERVE was examined in Arabidopsis. RNA (and DNA) was isolated from the leaves of 6 week old Arabidopsis thaliana plants using an RNeasy Plant Mini Kit (Qiagen) according to the manufacturer’s protocol.

The concentration of extracted nucleic acids was measured using a Nanodrop spectrophotometer and the quality monitored by formaldehyde agarose gel electrophoresis. The final preparation (0.36 µg/µl) contained DNA as well as total RNA and was treated with DNase to measure transcripts. 15 µl of the preparation was treated with 3 µl of RQ1 RNase-free DNAse (1 unit/µg RNA, Promega) in a 30 µl reaction after the addition of 3 µl of RQ11 RNase-free DNase 10x reaction buffer and 9 µl double-distilled water for 30 min at 37 °C. The reaction was stopped with 3 µl DNase stop solution (20 mM EGTA) and incubated at 65 °C for 10 min to denature the DNase prior to RTPCR.

PCR and RTPCR

The synthetic DNA oligonucleotides used for PCR and RTPCR of the Arabidopsis thaliana mitovirus NERVEs were CTTGCTCGCTTTGGCAGGAAG (5′ sequence for chromosomal copy), GCGGCGTTTGTTTGTAATCGGT (5′ sequence for the mitochondrial copy) and CAATGCACGATGCCATCGTTTGA (3′ sequence for both copies). The oligonucleotides used for the control (rbcL) sequence were TCAGGTGGACGAAAGTGTAAAG and GAACCACTCCCAGTTAGCATAG.

Pre-incubation mixes consisted of 10 µl of template (200 ng), 1 µl of 10 mM dNTPs, and 1.5 µl of each primer (10 µM). RTPCR reactions were 14 µl of pre-incubation mix, 5 µl of double distilled water, 5 µl of 5x Qiagen buffer, and 1 µl of Qiagen one-step RTPCR enzyme mix. DNA synthesis reactions consisted of one cycle at 50 °C for 30 min; one cycle of 95 °C for 10 min; 40 cycles of 94 °C for 15 s, 55 °C for 15 s, 72 °C for 30 s and a final cycle of 72 °C for 10 min followed by cooliing at 12 °C for 10 min and refrigeration at 4 °C. Analysis of products was by electrophoresis in 1.4% agarose gels run with 1 µg/ml ethidium bromide in 1XTAE.

Results and Discussion

Mitovirus NERVEs in mitochondrial genomes

The mitovirus NERVEs present in plants vary from nearly complete versions of the RdRP to remnants barely detectable in sequence searches (E values varying from 10−40 to 0.01). For simplification, we primarily limit our discussion to the mitovirus NERVEs present in completely sequenced mitochondrial genomes of plants, which currently derive from 90 different organisms. In these mitochondrial genomes, there are more than 175 partial or nearly complete copies of the mitovirus RdRP. In some cases, there is more than one mitovirus NERVE on a single mitochondrial DNA. In others, there is more than one mitochondrial genome per cell (e.g., Amborella), some of which have mitovirus NERVEs. Remarkably, an alignment of these 175 NERVEs shows that all the well-known conserved motifs (A–F) of the RdRP (Bruenn, 2003) are preserved in the paleovirus copies (Fig. 1, Figs. S1A and S1B). This is reminiscent of the perfect preservation of the totivirus conserved motifs in totivirus RdRP NERVEs (Taylor & Bruenn, 2009).

Figure 1 Consensus sequence of narnavirus NERVEs.

Consensus sequence of 175 mitovirus NERVEs and 29 mitovirus RdRPs and similarity plot along the sequence. FABCDE indicate the conserved motifs of RdRPs. Alignment was generated by MAFFT as described in Materials and Methods.

In many cases, there are mitovirus NERVEs both in mitochondrial and nuclear genomes and the direction of transfer is clear from an examination of the synteny of the miotochondrial and nuclear genomes in the region containing the mitovirus NERVE (Fig. 2), since genes of mitochondrial origin now appear in the nuclear genome. As is well known, blocks of mitochondrial DNA have been transferred to the nuclear genome (Leister & Kleine, 2011). In the cases shown, these blocks included mitovirus NERVEs. There are many mitovirus NERVEs in plant mitochondrial and/or nuclear genomes not included in our survey because the mitochondrial genomes of their plants have not been completely sequenced. For instance, there are at least 5 mitovirus NERVEs in the tomato genome, but its completed mitochondrial genome is not yet available. Consequently, we cannot say that every mitovirus NERVE present in the plant nuclear genome is derived from a mitochondrial copy, but where comparisons can be made this appears to be the case.

Figure 2 Synteny between mitochondrial and chromosomal genomes including regions with mitovirus NERVEs (indicated by NN) for three plant species.

(A) The Arabidopsis mitovirus NERVE used in the RTPCR experiment of Fig. 5. (B) Nicotinia benthamiana. (C) Vitis vinifera. Synteny diagrams were generated by CoGe, as described in Materials and Methods.

The phylogeny of plant mitochondria does not accurately represent accepted plant phylogeny, probably because of frequent horizontal transfer of mitochondrial genes (Xi et al., 2013). There is also very poor resolution among closely related species, which is a consequence of the great degree of sequence conservation in coding regions but complete lack of synteny among plant mitochondrial genomes (Palmer & Herbon, 1988). Given the rate of scrambling of mitochondrial genomes, the existence of synteny between mitochondrial and nuclear genomes within blocks of transferred mitochondrial DNA (Fig. 2) implies recent transfer events.

The plant mitochondria containing mitovirus NERVEs are all seed plants (Spermatophyta) or club mosses (Lycopodiophyta). Mitovirus NERVEs are missing from all of the Chlorophyta and from all of the Streptophyta except for the Embryophyta (higher plants) and within the higher plants are confined to the Tracheophyta (vascular plants). Among the Viridiplantae, there are sequences for mitochondria from Chlorokybophyceae (Chlorokybus atmophyticus), Mesostigmatophyceae (Mesostigma viride), Charophyceae (Chara vulgaris), Coleochaetophyceae (Chaetosphaeridium globosum), Anthocerotophyta (Megaceros aenigmaticus), Bryophyta (Anomodon attenuates), and Marchantiophyta (Marchantia polymorpha) and all are lacking mitovirus NERVEs (Table S1). Among the Tracheophyta (vascular plants), there are examples only from the Lycopodiophyta (Huperzia squarrosa) and Spermatophyta (all the other NERVE containing mitochondria) and mitovirus NERVEs are present in both. Our data are therefore consistent with an original integration of mitovirus cDNA in the mitochondria of the common ancestor of the vascular plants (Tracheophyta), as shown in the tree adapted from Davis, Xi & Mathews (2014), Fig. 3. However, we cannot exclude multiple integration events subsequent to the origin of vascular plants.

Figure 3 Abbreviated cladogram of the Viridiplantae according to Davis, Xi & Mathews (2014).

The plant groups in which mitovirus NERVEs have been demonstrated are indicated by black dots. All of the groups of vascular plants without demonstrated mitovirus NERVEs (e.g., ferns, Pinaceae, etc.) have no sequenced mitochondrial genomes.

Figure 4 Abbreviated phylogeny of the mitovirus NERVEs and the known narnaviruses.

The sequences of 61 plant mitochondrial mitovirus NERVEs of 100 amino acids or longer in length were aligned with the 31 known narnavirus RdRPs and two representative levivirus RdRPs and a phylogram rooted by the levivirus sequences generated as described in Materials and Methods. aLRT support values are shown. Several mitoviruses listed by name are ambiguous because more than one mitovirus is present. These are Ophistioma mitovirus 1c (AGT55876), Sclerotinia sclerotiorum mitovirus 7 (AHE13866) and Sclerotinia homeocarpa mitovirus (AAO21337).

The plant mitochondrial mitovirus NERVEs are all derived from one or more fungal mitovirus RdRP genes, as shown in the phylogram of Fig. 4. The phylogram was derived from an alignment of the 61 plant mitovirus NERVEs longer than 100 amino acids with all of the known Narnaviridae (31 viruses including the two Saccharomyces cerevisiae viruses) and rooted by the leviviruses (alignment is shown as Fig. S2). It places the mitovirus NERVEs clearly within the mitoviruses. All of the main branch support values are reasonable. Since all of the mitoviruses are present in plant pathogenic fungi, transfer of mitoviruses from their fungal hosts to plant mitochondria is not inconceivable.

There is a single prominent group of seed plants missing mitovirus NERVEs (the Poales, including Zea mays and Oryza sativa) (Table S1). Given the wide distribution of mitovirus NERVEs in the vascular plants, this is likely to have resulted from a loss rather than from multiple integration events. Actually, the vestiges of mitovirus NERVEs are still detectable in some of the grains, for instance in the Sorghum bicolor nuclear genome, with a region of chromosome 6 with similarity (E value 0.26) to the ABCDE portion of the mitovirus RdRP (Fig. S3).

Expression of mitovirus NERVEs

The widespread preservation of NERVEs (and their open reading frames) in eukaryotes suggests that they may have important functions. A prerequisite for function is expression. We chose to examine the expression of a single mitovirus NERVE in Arabidopsis whose mitochondrial and nuclear expression can be distinguished. Many of the mitovirus NERVEs preserve much or all of the RdRP amino acid sequence in long open reading frames. One of these is the NERVE in Arabidopsis, which has copies in both nuclear and mitochondrial genomes (Fig. 2). This is a region of 274 amino acids in the mitochondrial copy and 246 amino acids in the nuclear copy, both of which encompass the entire FABCDE conserved region of the RdRP and end at the same residue. Since these sequences are located at the extreme end of a syntenous region, it was possible to construct oligonucleotide primers in which one primer resides in the conserved RdRP region and the other in the unique, non-syntenous nuclear or mitochondrial DNA adjacent to the common region but possibly on the same transcriptional unit. Hence an RTPCR experiment can determine if one or both of these sequences are transcribed. Total RNA from Arabidopsis thaliana (with some contaminating DNA) was subject to RTPCR using these primers. The result (Fig. 5) shows that only the mitochondrial NERVE is transcribed. The conservation of the core RdRP sequence in a long open reading frame and its transcription are consistent with selection for expression of the NERVE protein, possibly as a means of interfering with mitovirus propagation, which, presumably, would occur in the mitochondrion. The mechanism of interfence might be at the protein level (Taylor et al., 2013; Taylor & Bruenn, 2009; Taylor et al., 2011; Taylor, Leach & Bruenn, 2010) or via RNA silencing.

Figure 5 RTPCR of Arabidopsis thaliana total nucleic acids.

Reactions before (A. DNA) or after (B. RNA) DNase treatment were tested for the nuclear or mitochondrial mitovirus NERVE or for the control (rbcL) sequence and analyzed on a 1.4% agarose gel as described in Materials and Methods. The first lane has DNA size markers, of which two are shown, one of 500 bp and one of 250 bp.

Conclusions

The existence of mitovirus NERVEs in plant mitochondria (Marienfeld et al., 1997) is remarkable for several reasons. First, the mitovirus NERVEs are clearly derived from fungal mitoviruses rather than from some hypothetical ancestral virus native to plant mitochondria. Given that the only encapsidated viruses to which the mitovirus are related are the leviviruses of bacteria, it is tempting to postulate that the mitoviruses of fungi are remnants of the original capture of bacteria by eukaryotes (the ancestors of mitochondria) and therefore predate the divergence of plants and animals (Koonin & Dolja, 2014). However, there are currently no known plant (or animal) Narnaviridae, and the monophyletic nature of the plant mitovirus NERVEs and their presence in only a subset of plants does not support this model for the origin of mitoviruses. Rather, it implies transfer from fungal mitochondria to plant mitochondria, as has been suggested previously (Marienfeld et al., 1997; Xu et al., 2015). Given the intimate association of plant pathogenic fungi with their hosts, this seems possible, especially given the known horizontal transfer of plant mitochondria within the plant kingdom (Leister & Kleine, 2011). Plant endophytic fungi are an even more likely origin (Bao & Roossinck, 2013). A similar origin for the plant endornaviruses by horizontal transfer of some segments from fungal endornaviruses has been proposed (Koonin & Dolja, 2014; Rastgou et al., 2009). Second, like all NERVEs, integration into DNA genomes requires reverse transcription. Where it is possible to trace the origin of a NERVE integration event, integration is the result of promiscuous substrate switching by a transposon reverse transcriptase (Ballinger, Bruenn & Taylor, 2012). Presumably this is the case for the mitovirus NERVEs. Third, the plant mitochondrial genetic code is the standard code, while the fungal mitochondrial genetic code uses UGA as a tryptophan codon, so the mitovirus NERVEs all must use the standard code. This implies the switching of all UGA codons to UGG either prior to the integration event or afterwards (Shackelton & Holmes, 2008). The switch could easily have happened prior to integration, so that a fungal mitovirus adapted to growth in plant mitochondria and the plant responded by integrating a copy of its RdRP in such a way as to interfere with viral replication. Fungal viruses with larger genomes than that of the mitoviruses are known to successfully adapt to hosts with alternate genetic codes (Taylor et al., 2013). In addition, the Saccharomyces cerevisiae narnaviruses have escaped from mitochondria, replicate in the cytoplasm, and consequently use the standard genetic code (Koonin & Dolja, 2014).

There are three methods by which we might corroborate a single integration of a mitovirus cDNA into a plant mitochondrial genome: correspondence of a mitovirus NERVE phylogeny with that of plant mitochondria; synteny of regions around mitovirus NERVE integration sites among plant mitochondrial genomes; or monophyly of mitovirus NERVEs. Unfortunately, a reliable mitovirus NERVE phylogeny is impossible to construct with current data, since the conserved regions of the RdRP do not vary significantly among the mitovirus NERVEs and the regions between them show very little sequence conservation. A much larger collection of mitovirus NERVEs would be required to construct a phylogeny with any confidence. In addition, there is not enough synteny among plant mitochondrial genomes to detect conservation around NERVE integration sites. It is possible that multiple integration events took place, but the monophyletic nature of the mitovirus NERVEs and their preservation of the functional motifs of the RdRP argues for a single integration event. If there was a single integration event, it had to take place prior to the origin of vascular plants in the early Silurian (Steemans et al., 2009), providing the earliest evidence for viruses of any kind, about 400 million years ago, older than the estimate of 310 million years ago for insect DNA viruses (Theze et al., 2011).

Supplemental Information

Figure S1A MAFFT alignment of 175 mitochondrial narnavirus NIRV RdRPs with those of 29 mitoviruses

A FASTA version of this alignment is available on request.

Click here for additional data file.

Figure S1B MAFFT alignment of 175 mitochondrial narnavirus NIRV RdRPs with those of 29 mitoviruses

A FASTA version of this alignment is available on request.

Click here for additional data file.

Table S1 Mitochondrial genomes analyzed for narnavirus sequences

A list of the plant mitochondrial genomes with complete sequences analyzed for narnavirus sequences. There are 40 sequences with narnavirus NERVEs and 50 without.

Click here for additional data file.

Figure S2 Alignment of the 61 narnavirus NIRV RdRPs of 100 amino acids or longer with those of the 29 mitoviruses, the two yeast narnaviruses, and two representative leviviruses

This is the alignment used to generate the cladogram of Fig. 4. A FASTA version of this alignment is available on request.

Click here for additional data file.

Figure S3 BLAST identification of a putative degenerate narnavirus NIRV in the sorghum mitochondrial DNA

Click here for additional data file.

We thank Charlotte Lindqvist, Jeffrey Boore, Matt Ballinger, and Riva Bruenn for comments on the manuscript.

Additional Information and Declarations

Competing Interests

Author Contributions

Jeremy Bruenn is an Academic Editor for PeerJ.

Jeremy A. Bruenn conceived and designed the experiments, performed the experiments, analyzed the data, contributed reagents/materials/analysis tools, wrote the paper, prepared figures and/or tables, reviewed drafts of the paper.

Benjamin E. Warner and Pradeep Yerramsetty conceived and designed the experiments, performed the experiments, analyzed the data, contributed reagents/materials/analysis tools, reviewed drafts of the paper.

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
