# Peer review of "Widespread mitovirus sequences in plant genomes"

_PeerJ, doi:10.7717/peerj.876_

## Round 0.1 · original submission · Minor Revisions

· Academic Editor

Minor Revisions

Please address all comments of both reviewers in your resubmission. Regarding figure 5, the image of the gel should have a black background instead of pink (in the file that was submitted). Please correct this, and also note in the legend that the first lane had molecular weight markers. That 250 and 500 refer to the markers in base pairs (bp) should also be noted.

·

Basic reporting

No Comments

Experimental design

No Comments

Validity of the findings

No Comments

Additional comments

The manuscript by Bruenn et al. reports the mitovirus-like sequences are widespread in plant genomes. The authors also claimed that mitovirus NERVES of plant were obtained from plant pathogenic via mechanism of horizontal gene transfer. Although the previous reports have been mentioned that mitovirus genomic sequence may transfer between fungi and plants (Marienfeld et al., 1997; Xu et al., 2005), the related research is more extensive and in-depth in the present manuscript and has some significant scientific value. Thus the manuscript contains some interesting novel information with solid data.
1, Based on the ICTV rules, family Narnaviridae is compose of two Genus: Mitovirus and Narnavirus. Narnavirus should refer two species: Saccharomyces 20S RNA narnavirus and Saccharomyces 20S RNA narnavirus, usually not refer mitovirus. In my opinion, narnavirus should change into mitovirus in the manuscript including title.
2, Why no fungal mitochondria genomes carrying mitovirus-related elements were discovered should be written and discussion.
3, The name of virus family should be italic (for example: line 3, 7, 9)
4, Fig. 5 are poor in quality

Reviewer 2 ·

Basic reporting

No comments

Experimental design

No comments

Validity of the findings

No comments

Additional comments

The paper by Bruenn and colleagues describes the comparison of narnavirus-related sequences mined from plant mitochondrial and nuclear genomes (narnavirus NERVEs) to each other and to narnavirus RNA sequences from fungi. It represents an important step in research examining the interface of plant and fungal viruses and defense response of those host organisms.

One interesting finding is that plant mitochondrial narnavirus NERVEs are monophyletic, supporting the hypothesis of a single entry from fungi to plants, followed by a reverse transcription and integration event into a plant mitochondrial genome and then expansion through much of the plant kingdom after that initial invasion. The other finding of interest is that the Arabidopsis mitochondrial narnavirus NERVEs are expressed as transcripts, while its nuclear counterpart is not. How this relates to function was not examined, although the authors conjectured that such mitochondrial RNA expression might prime host defense response.

A few suggestions for improvement:

1. The authors indirectly introduce the idea of RNA silencing playing a role in plant vs. fungal narnavirus sequence presence in lines 200-204. It would be interesting for them to explore this idea more explicitly. RNA silencing is more complex and thoroughly developed as a molecular mechanism in plants than it is in fungi, but is known to be an important antifungal defense response in both. This would seem like an interesting avenue for experimental investigation. The authors have the expertise and are in a good position to lay out some of the possibilities here.

2. The authors rightly mention plant pathogenic fungi as a likely source of plant mitochondrial narnaviral NERVEs, for example on lines 170-172, 219-221. In this context, they should also mention plant endophytic fungi as the possible original source.

3. The phylogeny shown in Figure 4 is confusing: at least one Ophiostoma and one Sclerotinia mitovirus are con-specific, so we need more detail to interpret the phylogeny. Perhaps this can be done most easily in the figure legend.

---

## Round 0.2 · accepted · Accept

· Academic Editor

Accept

I believe that the concerns of both reviewers have been adequately addressed in this revised version of the manuscript.